# Brief Communication: Nearly balanced glaciers in the Hunza catchment (Karakoram) since the 1970s

Tobias Bolch[1,2], Tino Pieczonka[2], Kriti Mukherjee[2], Joseph Shea[3,4]

[1]Department of Geography, University of Zurich, 8057 Zurich, Switzerland
[2]Institute for Cartography, Technische Universität Dresden, 01069 Dresden, Germany
[3]International Centre for Mountain Development (ICIMOD), Kathmandu, Nepal
[4]Center for Hydrology, University of Saskatchewan, Saskatoon, Canada

*Correspondence to*: Tobias Bolch (tobias.bolch@geo.uzh.ch)

**Abstract.** Previous geodetic estimates of mass changes in the Karakoram revealed balanced budgets or a possible slight mass gain since the year ~2000. Indications for longer-term stability exist but only very few mass budget analyses are available before 2000. Here, we show based on 1973 Hexagon KH9, ~2009 ASTER, and the SRTM DTM that glaciers in the Hunza River basin (Central Karakoram) were on average in balance or showed slight insignificant mass loss within the period ~1973 – 2009. Heterogeneous behaviour and frequent surge activities were also characteristic for the period before
2000. Surge-type and non-surge-type glaciers showed on average no significantly different mass change values. However, some individual glacier mass change rates differed significantly for the periods before and after ~2000.

## 1 Introduction

Glacier melt water is of high importance for the run-off of the Indus River (Immerzeel et al., 2010) but the exact glacier share is not known. This is partly due to the lack of knowledge about precipitation, snow cover, and snow water equivalent,
but also about glacier mass balance, their characteristics and their responses to climate change. Karakoram glaciers, which occupy a large portion of the glacierized area of the Indus basin, have recently shown unusual behaviour: on average no significant area changes but frequent advances and surge activities have been observed during the last decades (Bhambri et al., 2013; Bolch et al., 2012; Copland et al., 2011; Hewitt, 2011; Rankl et al. 2014). Geodetic mass estimates revealed balanced glacier mass budgets or even slight mass gain since ~2000 (Gardelle et al., 2013; Kääb et al., 2015; Rankl and
Braun, 2016). However, mass budget analyses for Karakoram glaciers prior to the year 2000 are rare. While this paper was in open discussion, Zhou et al. (2017) reported a Karakoram-wide mass budget of $-0.09 \pm 0.03$ m w.e. a$^{-1}$ for the period ~1970 to 2000. For Siachen Glacier in eastern Karakoram, Zaman and Liu (2015) corrected the erroneous value of -0.51 m w.e. a$^{-1}$ given by Bhutiyani (1999), and estimated the mass budget for the period 1986-1991 to be between +0.22 m and -0.23 m w.e. a$^{-1}$. Agarwal et al. (2016) reported a mass budget of $-0.03 \pm 0.21$ m w.e. a$^{-1}$ of Siachen Glacier for the period 1999
and 2007. Herreid et al. (2015) found no significant change in debris-coverage of the glaciers in the Hispar and Shimshal

sub-regions of the Hunza River basin for the period 1977 until 2014 and concluded that this might be due to balanced glacier budgets during this period. Temperature measurements that are available since 1961 in the Karakoram show, in contrast to many other regions of the globe, a consistent decline in summer and an increase during winter (Fowler and Archer, 2006). Hence, these measurements would support the assumption that glaciers would be in balance or slightly positive conditions over the last several decades of the 20[th] century.

Declassified stereo satellite images from the 1960s and 1970s such as Corona KH-4 and Hexagon KH-9 have been proven to be suitable to generate digital terrain models (DTMs) and assess glacier mass changes since the 1960s (Bolch et al., 2008; Pieczonka et al., 2013, Maurer et al., 2016). Hence, the aim of this study is to revisit existing information and extend the time series back to some of the earliest available satellite imagery. We focus on the Hunza catchment in the Central Karakoram (Figure 1) where high heterogeneity of glacier behaviour was found in previous studies (e.g. Bolch et al., 2012; Quincey and Luckman, 2014). Moreover, suitable Hexagon KH-9 data from the 1970s and recent stereo data from ~2009 such as ASTER and Cartosat-1 data were available. The Hunza River is a tributary to the Gilgit River, which flows into the upper Indus. The area of the basin is about 13,715 km² and approximately 25% of the basin is covered by glaciers. These glaciers constitute more than 15% of the glacierized area of the entire Karakoram. Frequent surges reported for several glaciers in this basin (Quincey and Luckman, 2014; Copland et al., 2011; Rankl et al., 2014) complicate the analysis of mass budgets as often only a certain part of a surge is captured.

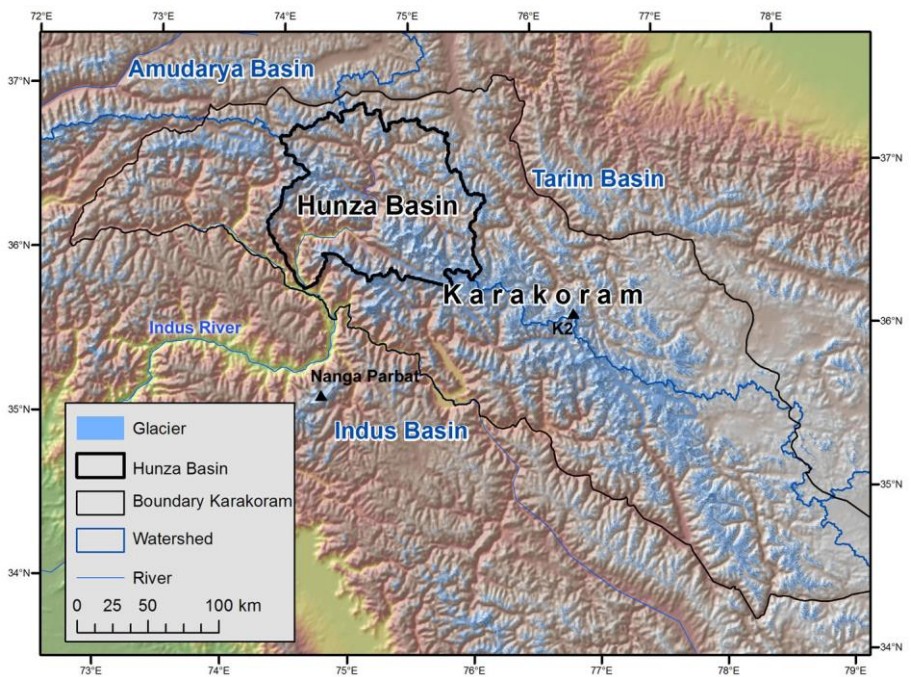

**Figure 1: Overview map of the study region.**

## 2. Data and Methods

### 2.1 Data

The SRTM digital terrain model (DTM) version 4, with a spatial resolution of 1 arc second (~30 m, SRTM1) was utilized as reference dataset. The SRTM1 DTM was acquired by the use of two C-Band radar antennas (operating in interferometric mode) during 11 - 22 February 2000 and is frequently used for glaciological investigations. It can be assumed that the represented ice surface of the ablation region is close to the surface at the end of the 1999 ablation period, assuming a full penetration of the radar beam into snow (Paul and Haeberli, 2008). However, deeper radar penetration can be expected in the accumulation region (Berthier et al., 2006). Data voids which are mainly restricted to some accumulation areas were filled with ASTER GDEM2 data (see section 3.2). This DTM is a merge of several ASTER scenes covering the period ~2000 – 2010 (Slater et al., 2011). We used ten on demand generated ASTER DTMs (product AST14DEM) from five different acquisition dates of the year 2008 – 2010. A small missing strip was filled with a DTM from 2013 scenes (Table S1, Fig. S1). ASTER scenes were visually checked and from the most promising available (close to end of ablation season, minimum snow cover, no clouds) the respective DTMs were ordered and used for DTM differencing.

Two high-resolution Cartosat-1 stereo scenes captured on 11 July 2010 (Table 1) were used to compare and investigate the consistency of the results obtained with the lower resolution ASTER DTM. Cartosat-1 (IRS-P5) was launched by Indian Space Research Organisation (ISRO) in May 2005. The satellite has two high resolution (2.5 m) panchromatic sensors recording stereo images along the track (Titarov, 2008). The major advantage of this dataset, besides the high spatial resolution, is the 12 bit radiometric resolution. Unfortunately, the spatial coverage is relatively small (25 x 25 km) and our two stereo pairs cover only one large glacier (Khurdopin Glacier, glacier nr. 6 in Figure 2) in full.

Declassified Hexagon KH-9 imagery which has a ground resolution of about 8 m and a coverage of about 250 x 125 km allowed us to extend the time series back to 1973 (Table S1). The KH9-Hexagon mission was part of the US keyhole reconnaissance satellite program whose images were declassified in 2002. Imagery from this program have already been applied to investigate glacier mass changes (e.g. Pieczonka et al., 2013).

The ICIMOD glacier inventory (Bajracharya and Shrestha, 2011), also available through the GLIMS database (www.glims.org), was used as a baseline data set and was manually adjusted based on the utilized optical imageries for DTM generation and Landsat ETM+ scenes of the years 2000 and 2001.

### 2.2 DTM generation, postprocessing, differencing and uncertainty

All KH-9 DTMs were generated with Erdas Imagine 2014 Photogrammetry Suite using the frame camera model with a focal length of 30.5 cm. Image pre-processing includes the elimination of internal distortions based on the regularly distributed réseau crosses (originally included to correct film distortion effects) and their removal thereafter, following Pieczonka et al. (2013). GCPs were collected from Landsat 7 ETM+ imagery with SRTM1 as a vertical reference (Pieczonka et al., 2013).

Though GCP collection in rough terrain is challenging, we were able to identify 26/28 GCPs (Table S2) located at mountain peaks, large terrain features, and bridges. GCPs were distributed throughout the scenes and at different elevations.

Fiducials were measured manually considering the principal point in the image centre. All stereo images have been processed with a RMS of <~1.5 pixels (Table S2). The final Hexagon KH-9 DTMs cover the entire Hunza basin. A small gap of about 20 pixels exists between the two generated DTMs.

The Cartosat-1 stereo pairs have also been processed using PCI Orthoengine 2014 with 32 and 35 GCPs. To improve the quality of the DTMs, image enhancement techniques were applied prior to DTM generation in order to overcome low image contrast and temporal differences in image acquisition. The root mean squared error (RSME) varied between 0.3 and 3.9 pixels (Table S2). The spatial resolution of all DTMs was chosen as 30 m.

In order to obtain reliable results on glacier surface elevation changes, the DTMs must be properly co-registered (Nuth and Kääb, 2011). As we observed tilts when differencing the original DTMs, we first minimized elevation differences between the different DTMs with respect to the SRTM1 master DTM by applying a first order trend correction. We considered only elevation differences ($\Delta h$) between ±150m over non-glacierized terrain with slopes less than 15° (Bolch et al., 2008; Pieczonka et al., 2013). Subsequently, all DTMs were further co-registered following the approach by Nuth and Kääb (2011). The final horizontal shift between all DTMs and SRTM1 were less than or equal to one pixel (≤30 m) on average.

To calculate surface elevation changes, we subtract each older DTM from more recent DTM, and mosaic the difference grids to facilitate processing. Where the ASTER DTMs from different time periods overlapped, we calculate a weighted mean elevation change based on the time of acquisition and glacier coverage (Figure S1, Table S3).

Data voids and mismatches that result in incorrect elevation values can occur in areas with low image contrast such as cast shadows and bright snow. Mismatches due to snow in the accumulation regions led to unrealistic low elevation values using KH-9 data that would subsequently lead to unrealistic surface lowering values in parts of the accumulation region (Pieczonka and Bolch, 2015). However, thickness change distributions for glaciers with negative mass budgets typically have a minimum lowering at the glacier head with increasing values towards the glacier front following a non-linear trend (Huss et al., 2010). This pattern is different for surging glaciers that often exhibit high positive $\Delta h$ values at comparatively low elevations, strong surface lowering around the ELA, and then decreasing $\Delta h$ values towards the upper reaches (Gardelle et al. 2013, Rankl and Braun, 2016). Elevation change patterns can also be affected by thick debris cover where the highest lowering usually does not occur close to the terminus but upstream (Bolch et al., 2008).

As both debris-covered glaciers and surge-type glaciers are common in the investigated region we could not apply a general threshold to remove $\Delta h$ outliers, but used the general assumption that lower elevations show stronger $\Delta h$ variability than higher elevations. This should still be true for surging glaciers and those with balanced conditions. The related calculations followed the approach by Pieczonka and Bolch (2015) and used a sigmoid function allowing a larger range of $\Delta h$ values in the middle part of the ablation region to preserve the signal of surging glaciers and a narrower range ($-20 \leq \Delta h \leq 20$ m) at the glacier head. We filled all data gaps (including the gap between the two KH9 DTMs) by means of ordinary kriging in order to get the weighted moving average based on neighbouring pixel values.

The penetration of the radar beams into firn and snow has to be considered in case of the comparison of DTMs generated from microwave data such as the SRTM1. However, the value can only be estimated as it depends on several unknown parameters (e.g. snow depth and characteristics) and is therefore one major source of uncertainty (Kääb et al., 2015; Gardelle et al., 2013). We applied a penetration correction of 2.4 ±1.4 m as a mean average. This value was suggested by Kääb et al.

(2012), who analysed the beam penetration of the C-band SRTM data in a similar region of the Karakoram. The conversion of volume to mass changes needs to consider the combined ice and snow density. As both are unknown, we used a density of 850 ±60 kg/m³ as a reasonable and widely used assumption for a longer time period (Huss, 2013).

There is no best method to estimate the uncertainty (e) of the DTM differencing when no precise and well distributed GCPs are available. The standard deviation of the non-glacierized terrain can serve as a first estimate of the uncertainty and is 22m

for the difference between the KH9 and the SRTM DTM, 24 m (SRTM - ASTER DTM), and 26 m (KH9 - ASTER DTM). However, the standard deviation can be significantly higher than the real uncertainty as the spatial correlation is not considered (Rolstad et al., 2009). Therefore, we followed the widely applied approach developed by Gardelle et al., (2013):

$$E_{\Delta h} = \frac{E_{\Delta h_i}}{\sqrt{N_{eff}}} \qquad (1)$$

where $E\Delta h_i$ is the standard deviation of the mean elevation change of the non-glacierized terrain of each altitude band and $N_{eff}$ is the effective number of observations. The latter is calculated using the total number of observations $N_{tot}$, the pixel size $R$ (30 m), and $d$, the distance of spatial autocorrelation of the elevation change maps (1025 m)

$$N_{eff} = \frac{N_{tot} \cdot R}{2d} \qquad (2)$$

The overall uncertainty of the DTM difference is the average of $E_{\Delta h}$ weighted by the glacier hypsometry. A further uncertainty to be considered is the uncertainty of the mapped area of the glaciers $E_a$. We assume an uncertainty of 5% which is towards the upper bound of published estimates of the uncertainty of mapped glaciers based on similar satellite data (e.g.

Paul et al. 2013) taken into consideration that the delineation of debris-covered and avalanche-fed glaciers which are both common in the study region is more difficult. The final uncertainty ($E$) is calculated considering also the uncertainty of the radar penetration ($E_p$, ±1.4 m) and of the volume to mass conversion ($E_m$, ±7% of the elevation change):

$$E = E_{\Delta h} + E_a + E_p + E_m \qquad (3)$$


We did not apply a seasonality correction as most of the images were acquired close to the end of the ablation period but assume the effect is well within the considered uncertainty.

## 3. Results and Discussion

### 3.1 Glacier volume and mass changes, surge-type glaciers

Results for the period 1999 to ~2009 show heterogeneous glacier behaviour, with several surging glaciers in the study region (Figure 2). A northern tributary of Hispar Glacier thickened by approximately 150 m at the confluence of the glaciers. Khurdopin and its neighbouring glaciers in south-eastern Shimshal Valley both show significant thickening and thinning within their tongues which is typical of a surge occurring during the study period (Figure 2). In contrast, the large debris-covered Batura Glacier west of Shimshal Valley showed surface lowering throughout the tongue leading to an overall volume loss (Figures 2, 3). For the entire study region we found no significant mass changes (Table 1) which is in line with previous results for the period after 1999 in Central Karakoram (e.g. Gardelle et al., 2013). In addition, we add information for glaciers east of their study region (e.g. Batura Glacier) and cover the entire Hispar Glacier, one of the largest glaciers (length 50 km) in the Karakoram. For this region we cannot confirm positive mass budgets but our study indicates a slight mass loss similar to Rankl and Braun (2016) and Kääb et al. (2015). However, our uncertainties are larger due to the utilization of the lower resolution ASTER DTM. In addition, we emphasize that the analysed glaciers and period slightly differ which could be a reason for the (not significant) differences to the existing studies.

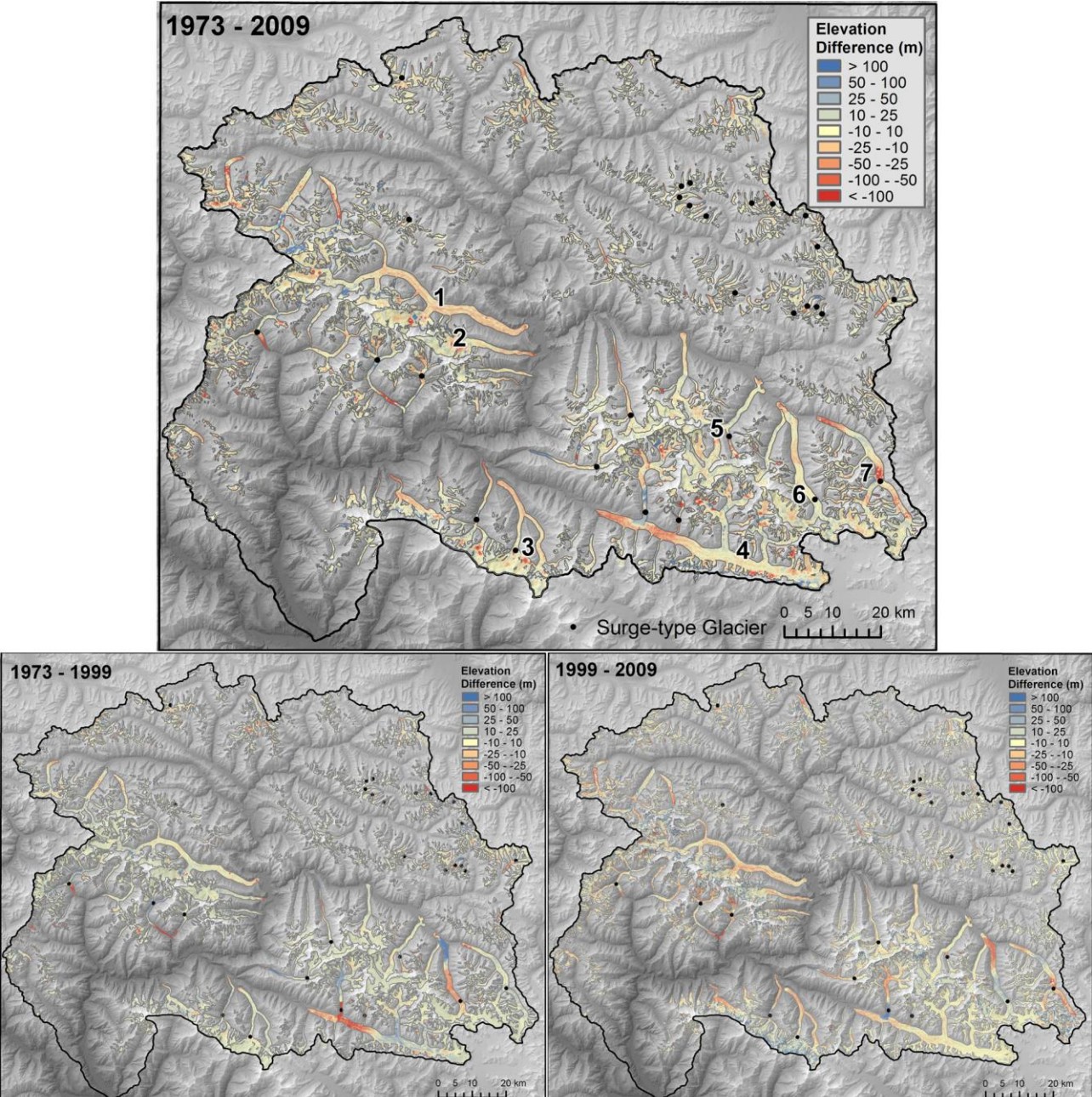

**Figure 2: Elevation difference between the ASTER and Hexagon KH9 DTMs (above), the Hexagon KH-9 and the SRTM1 DTMs (below, left) and the SRTM1 and ASTER DTMs (below, right). Black dots indicate surge-type glaciers, the numbers selected larger glaciers: 1: Batura, 2: Pasu, 3: Barpu, 4: Hispar, 5: Yazghil, 6: Khurdopin, 7: Vijerab**

Our extended time series indicates on average balanced budgets of the glaciers in Hunza Valley for the period 1973 to 1999 and slight, but insignificant mass loss for 1999 to ~2009. Heterogeneous behaviour was found in both periods (Table 1).

Differencing of the 1973 KH9 and ~2009 ASTER DTMs confirm the results of the individual periods without the uncertain penetration correction and reveal slight, insignificant mass loss during at least the last 40 years. Longer-term balanced budgets had been assumed previously based on existing information about (insignificant) area changes since the 1970s (Bhambri et al., 2013), on average similar debris coverage since 1977 (Herreid et al., 2015). Our results, based on Hexagon KH-9, SRTM and ASTER data, confirm previous studies and also the work of Zhou et al. (2017) who found on average balanced-budgets or only a slight mass loss for Central Karakoram between 1973 and 2000.

Most glaciers experienced similar mass budgets for both investigation periods. However, it seems that some glaciers had more negative budgets post-2000. This is especially true for the debris-covered Batura Glacier whose tongue showed significant lowering during 1999 – 2009. Over the entire study period there is no significant difference in the mass budgets of surge-type and non-surge-type glaciers, a result also found by Gardelle et al. (2013) and surge-type glaciers also showed more negative values in the recent period. Kurdophin Glacier, for example, experienced a significant thickening near the snout and a significant lowering around the ELA, both combined resulting in an about zero mass budget for 1973 and 1999 and significant lowering at the lower part of the tongue leading probably to a mass loss for 1999 – 2009 (Figures 2, 3).

For the recent period, the westernmost tributary of Hispar Glacier shows a clear sign of surging with significant elevation gain of more than 100 m at the confluence with the main glacier. This tributary also clearly thickened in the middle reaches during the earlier period (Figure 2). These elevation change characteristics are likely due to the fact that the ice builds up in the reservoir area during the quiescent phase and the active surge event transfers ice mass to lower elevations where it is then more prone to melting. The very high surface lowering rates observed in the middle part of the tongue of Hispar Glacier hints also to a past surge event. It is therefore important to cover the entire surge cycle of surge-type glaciers in order to have valid estimates of their mass budgets. This is the case for most of the glaciers in our study region as we cover a period of almost 40 years and the surge periodicity in the Karakoram is rather short with averages between ~25 and 40 years (Copland et al. 2011). Overall, we identified 28 surge-type glaciers (including 5 tributaries) based on the DTM differencing results in combination with morphological features like looped moraines or heavily crevassed glacier surfaces (Figure 2).

**Table 1: Glacier mass budget for different periods and glacier types**

| Nr. | Glacier Name | Glacier Type | Area (km²) | Glacier Mass Budget (m w.e. a⁻¹) | | |
|-----|--------------|--------------|------------|-------------|-------------|-------------|
| | | | | 1973 - 1999 | 1999 – ca. 2009 | 1973 – ca. 2009 |
| 1 | Batura | Debris covered | 236 | $0.00 \pm 0.10$ | $-0.39 \pm 0.26$ | $-0.12 \pm 0.09$ |
| 2 | Pasu | Debris free | 51 | $+0.05 \pm 0.11$ | $-0.13 \pm 0.26$ | $-0.09 \pm 0.10$ |
| 3 | Barpu | Surge type | 90 | $+0.03 \pm 0.08$ | $-0.10 \pm 0.18$ | $-0.15 \pm 0.08$ |

| | | | | | |
|---|---|---|---|---|---|
| 4 | Hispar | Surge-type Debris covered | 345 | -0.10 ± 0.08 | -0.11 ± 0.21 | -0.14 ± 0.08 |
| 5 | Yazghil | Debris covered | 99 | -0.02 ± 0.13 | -0.04 ± 0.32 | -0.01 ± 0.12 |
| 6 | Khurdopin | Surge-type Debris covered | 115 | -0.05 ± 0.09 | -0.14 ± 0.22 | -0.05 ± 0.08 |
| 7 | Vijerab | Surge-type Debris covered | 113 | +0.03 ± 0.10 | -0.31 ± 0.25 | -0.22 ± 0.09 |
| | **Whole region** | | **2868** | **-0.01±0.09** | **-0.08 ± 0.21** | -0.06 ± 0.08 |
| | Whole region | Non-surge-type | 2237 | 0.00 ± 0.08 | -0.03 ± 0.22 | -0.03± 0.08 |
| | Whole region | Surge-type | 631 | -0.03 ± 0.10 | -0.15 ± 0.30 | -0.09 ± 0.09 |

## 3.2    DTM generation and sources of uncertainty

Declassified KH-9 Hexagon data have been proven to be valuable for assessing geodetic glacier mass budgets (Pieczonka and Bolch, 2015; Pieczonka et al., 2013, Maurer et al. 2016). The main challenges in obtaining accurate results are

miscorrelations in the accumulation regions of glaciers and significant tilts and shifts of elevation trends making careful co-registration and post-processing necessary. This leads to higher uncertainties compared to more recent data with a similar spatial resolution. The glacier volume changes calculated based on the automatically derived ASTER DTMs (AST14DEM) and the SRTM DTM were similar to those using better quality higher resolution SPOT5 DTMs for a similar region (Gardelle et al., 2013). Off-glacier DTM differences show good agreement in general, but also regions with higher deviations where

the quality of the ASTER DTMs were lower (e.g. the western part of the study areas, Fig. S1). We found no significant difference between the mass budget results of Khurdopin Glacier calculated using Cartosat-1 data and the values calculated using the ASTER data (-0.14 ± 0.21 for 1999 – 2009 vs. -0.16 ± 0.13 for 1999 – 2010). This gives confidence that the ASTER DTMs which can be freely obtained for scientific purposes are of high value to calculate glacier volume and mass changes over a longer period of time as also shown for other regions (e.g. Berthier et al., 2016).

A further source of uncertainty are the data voids in the original SRTM data and voids due to the outlier filtering. About 20% of the total glacierized area for all analysed periods were identified as outliers and filled afterward by kriging interpolation. The voids are almost entirely located in the accumulation region where surface elevation changes are relatively small (e.g. Schwitter & Raymond, 1993) and where we restricted the maximum possible deviation. To assess the influence of the filling on the result we calculated the elevation change value for the whole basin and the period 1973 – 2009 (a) without void filling

(result: mean elevation difference -3.80 m or -0.09 m w.e. a$^{-1}$), (b) filling with zero (-2.98 m or -0.07 m w.e. a$^{-1}$), or (c) our applied interpolation method (-2.39 m or -0.06 m w.e. a$^{-1}$). The results show that average elevation change rates do not change significantly and the deviations are well within the uncertainty. About 10% of the glacierized area in our study region

is affected by data voids in the original SRTM data and previous studies showed that there can be significant deviations to reality in the data used to fill the voids (e.g. Kääb et al. 2012). In addition, the time stamp of the data is often unknown. We compared therefore the results of the void filled and the non-void filled version. Using the latter we calculated the surface elevation change for the existing pixels only. The resultant value of the mean surface elevation change for both periods differs only by about 0.02 m a$^{-1}$, which is also well within the estimated uncertainty. The minor differences are, besides the possibility that the ASTER GDEM data used for void filling are quite reliable for the study region, due to the fact that most of the voids are located in the accumulation regions where we restricted the maximum possible deviation. Only one larger glacier is also affected in the ablation region. This is Shishpar Glacier, a southerly facing exposed glacier, located south of Batura Glacier. The void-filled data allowed for detection of the surge activity between 1973 and 1999 with an estimated mass budget of +0.04 ± 0.19 m w.e. a$^{-1}$.

One of the major sources of uncertainty is the penetration of the radar beam into snow and ice when using the SRTM DTM. Gardelle et al. (2013) estimated a mean penetration of 3.4 m with values up to more than 9 m in the accumulation region. This value is higher than the 2.4 m we applied here following Kääb et al. (2012). However, applying the higher penetration value would only lead to a slight difference in mass change of ±0.03 m w.e. a$^{-1}$. The average surface elevation change without considering any radar penetration is -0.10 m a-$^{1}$ for the period 1973 – 1999 and +0.15 m a$^{-1}$ for the period 1999 – 2009. However, penetration is not an issue for calculation of the geodetic mass budget over the entire period (~1973 – 2009) as this uses optical data only. As the results of the individual periods are in good agreement with the values for the entire period, we are confident in the reliability of all mass budget calculations.

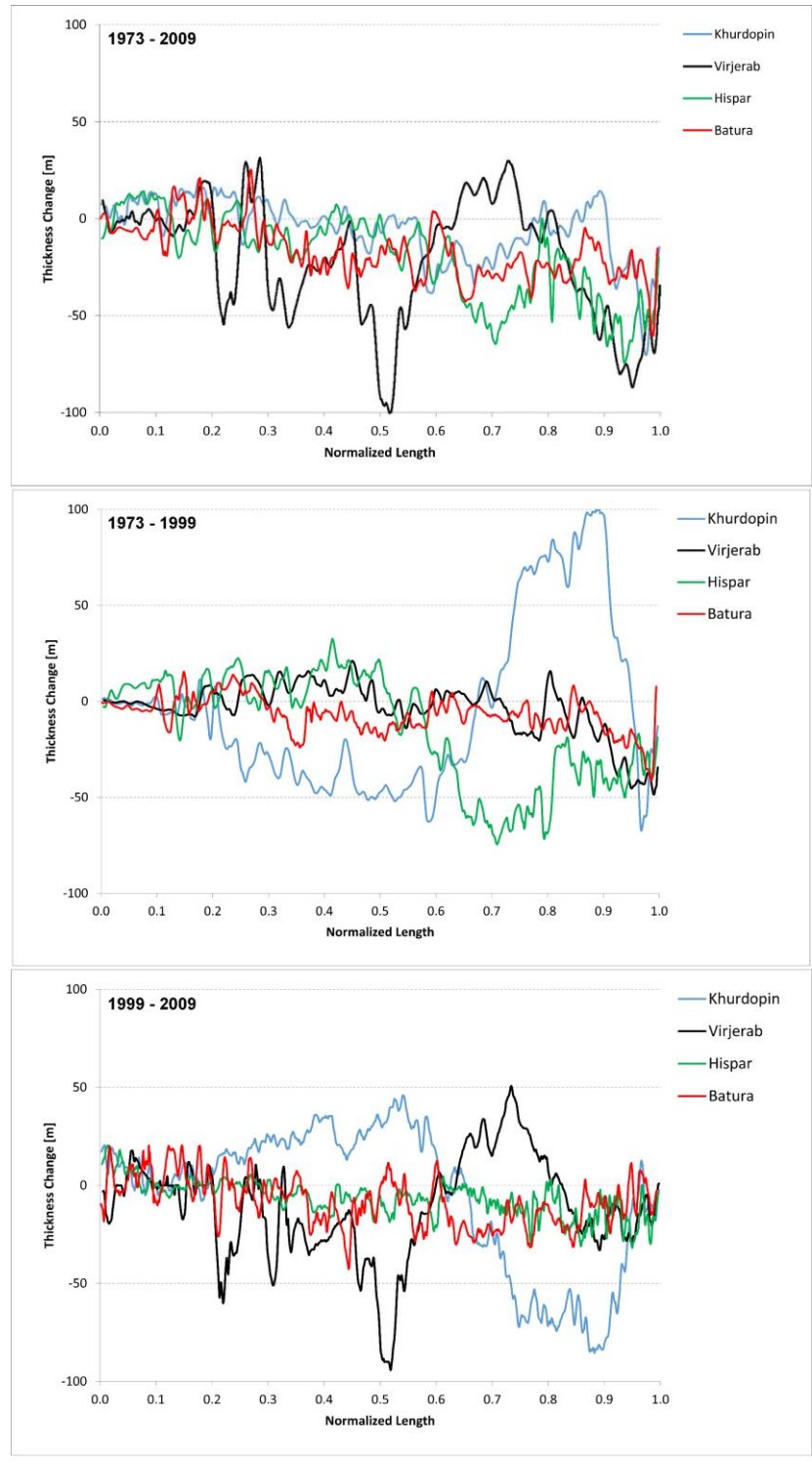

**Figure 3: Longitudinal profiles of surface elevation changes for selected glaciers for the entire period (1973 – 2009) and the subperiods 1973 – 1999 and 1999 – 2009 periods.**

## 4        Conclusion

We showed based on 1973 Hexagon, SRTM and ~2009 ASTER DTM data that heterogeneous behaviour and close to balanced-budgets in the investigated region in Hunza Valley (Karakoram) are not a recent phenomenon. Since at least the 1970s, glaciers in the study region have experienced on average only a slight, insignificant overall mass loss. However, significant differences can exist for individual glaciers for the two investigated periods 1973 – 1999 and 1999 – 2009. Especially surge-type glaciers show different elevation change characteristics and more negative mass budgets in the second period. During a surge, ice is transported rapidly from an upper reservoir zone to the ablation region and is prone to melt at the lower elevations. With the almost 40-year time-period considered here we show that the overall mass change of surge-type glaciers and non-surge-type are not significantly different when the entire surge cycle is considered. However, further long-term mass budgets studies are needed to confirm these findings.

**Author Contributions**

T.B. designed the study, performed all analysis, generated the figures and wrote the draft of the manuscript. K.M. generated the raw Hexagon and Cartosat-1 DTMs and co-registered the data. T.B. and T.P. supported the generation of the DTMs and the co-registration. J.S. contributed to the study design, and all authors contributed to the final form of the article.

**Acknowledgements**

This study was performed within the Cryosphere Initiative of the International Center for Integrated Mountain Development (ICIMOD), with the targeted support of the UK Department for International Development (DFID). ICIMOD is funded in part by the governments of Afghanistan, Bangladesh, Bhutan, China, India, Myanmar, Nepal, and Pakistan, and the Cryosphere Initiative is funded by the Norwegian Ministry of Foreign affairs. The views expressed are those of the authors and do not necessarily reflect their organizations or funding institutions. T. Bolch acknowledges funding through the ESA project Glaciers_cci (4000109873/14/I-NB), Deutsche Forschungsgemeinschaft (DFG) and The University of Colorado CHARIS project (Contribution to High Asia Runoff from Ice and Snow – (nsidc.org/charis) funded by the United States Agency for International Development. We thank F. Paul, two anonymous reviewers and the scientific editor for their constructive comments on the manuscript.

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
