# Peer review of "Brief Communication: Glaciers in the Hunza catchment (Karakoram) are nearly in balance since the 1970s"

_The Cryosphere, 2016_

## Referee Comment (RC1) · Anonymous Referee #1 · 14 Oct 2016

General comments:

This study utilizes digital elevation models extracted from satellite imagery (historical Hexagon and modern SRTM and ASTER) to calculate a regional geodetic mass balance for glaciers in the Hunza River basin, Karakoram region, using DEM differencing. The authors show that given the uncertainties of the methodology, the regional geodetic mass balance is not statistically different from zero change, consistent with previous mass balance studies on shorter more recent timescales. Their results suggest that the so-called "Karakoram anomaly" is not limited to the past ~15 years, but extends back to at least 1973. This is the first study using elevation differences to confirm this finding over a several-decade timespan, which supports previous studies showing no significant changes in debris cover or glacier area in the Karakoram over similar (1970's-present) time periods. Overall, it is a nice paper, and is ready for publication

after a few minor additions.

A table of values showing the standard deviation of mean elevation change between the ASTER and SRTM DEMs for assumed stable (non-glacier) terrain is needed to better assess the relative vertical accuracy of the DEMs.

Regarding the satellite imagery datasets, a paragraph, table, or figure to clearly show which DEMs are being subtracted from one another for each given time period, i.e. SRTM minus Hexagon for 1973-1999, and ASTER minus SRTM for 1999-2009. This would serve the clarify the methods section significantly.

Since the primary motivation of the paper is to extend the geodetic mass balance record further back in time, I would recommend an additional calculation of the full timespan (1973 - 2009) mass balance. This would also serve to validate the 1973-1999 and 1999-2009 mass balances, and remove the significant uncertainty regarding SRTM penetration into the ice.

The equations used for estimating uncertainty lean toward the more conservative side (i.e. large error bars). For example, linearly adding up the errors in Eq. 3 instead of adding in quadrature, which assumes that the error components in Eq. 3 are completely correlated with one another. The authors should make clear in the conclusion of the manuscript - results show no statistical difference from zero change, given the somewhat large/conservative uncertainties used with the DEM differencing method.

Specific comments:

P3 L6 Were any glaciers covered only partially by scenes from different years? If so, it may be best to use a weighted mean (weighted by the percentage of a glacier's area covered by each scene).

P3 L8 It is still somewhat unclear to me how the Cartosat-1 data is being used. I assume the authors compute Cartosat minus SRTM, then compare to ASTER minus SRTM in order to check consistency between the datasets. This should be further

clarified in the text.

P2 L18 "assuming a full penetration of the radar beam into snow..." - regarding the ablation region. What about additional penetration into the ice itself, is this taken into account?

P2 L20 It would be useful here to refer to the later section (3.2) so the reader can easily find the discussion regarding void filling with the ASTER GDEM2 and associated uncertainties.

P3 L24 "All stereo images have been processed with a RMS of $< \sim 1.5$ pixels." Which aspect of the stereo photogrammetry is this referring to? Is this the reprojection error of triangulated ground control points after bundle adjustment, or something to do with the reseau grid distortion removal, or something else? A more detailed explanation is needed to interpret the meaning.

P3 L28 See previous comment regarding P3 L24

P3 L32 What kind of spatial trend corrections were made? Rotation, translation, or perhaps polynomial surface corrections... if so are they first order (linear), or higher order polynomials, or some other method?

P4 L12 Was the outlier threshold applied to both Hexagon and ASTER data, or to Hexagon only? If no outlier filtering was needed for the ASTER DEMs, this should be stated explicitly in the text.

P4 L17 It would be helpful to know the percentage of total pixels excluded (using the outlier threshold filter) for each glacier, to ensure that no large regions were interpolated using the ordinary kriging; otherwise unrealistic elevations could result. The text later discusses the percentage of voids in the SRTM data, but says nothing regarding the percentage of data gaps in the Hexagon data.

P6 L11 "we confirm for the first time using elevation differences..." Should specify: over this longer time period (because elevation differences have been used over shorter

time periods in previous studies).

P6 L13 Going back to the previous comments regarding P4 L12 and P4 L17 - Since both time periods use different data sources and therefore contain differing amounts of data voids - a percentage of voids for each would help eliminate doubts regarding direct comparisons between the two time periods, which use SRTM - Hexagon for 1973-1999, and ASTER - SRTM for 1999-2009. Could a difference in data gaps/holes make a significant difference when comparing these datasets/time periods?

P6 L17 What is meant by "different surge stages in the two periods..."? What is different between the two periods regarding surges, the magnitude, timing, or something else? Or is the word "different" simply being used in a fashion equivalent to "separate"?

P8 L1 When calculating the mass budget with the non-void-filled version of SRTM for comparison, were the voids interpolated, or was the mass balance computed using only the volume change of existing pixels, then divided by the glacier area only covered by existing pixels? More details would be helpful.

Technical corrections:

P2 L7 complicate the

P3 L10 "The major advantage of this dataset is besides the high spatial resolution and also the 12 bit pixel depth" – strange wording. Should change to something like: "The major advantages of this dataset are the high spatial resolution and 12-bit pixel depth."

P7 L11 "... ASTER DEMs where lower..." change "where" to "were"

P8 L6 voids

P8 L4 versions

---

## Referee Comment (RC2) · Anonymous Referee #2 · 4 Nov 2016

This is a useful study that provides the first long-term information (since the 1970s) about geodetic glacier mass balance for a region of the Karakoram. This is a region where many recent papers have suggested that glaciers are changing little in mass, but prior to this study little previous information has been available about glacier elevation changes before ∼2000. The techniques are well described, the errors are well quantified, and useful final conclusions are produced.

Most of my comments are relatively minor and focused on technical issues, but there are two useful analyses that could be undertaken that would help to strengthen the paper:

1. A computation of the total elevation changes over the period 1973-2009 should be completed. This would help to validate the patterns shown in the individual periods,

potentially reduce the effect from individual surges, and provide evidence that mass balance has been stable over the long term.

2. Provide a plot and discussion of the change in geodetic mass balance with altitude for non surge-type glaciers. This could provide insight into whether changes are occurring at particular altitudes, even though the overall mass balance may be close to zero.

Individual comments:

P2, L22: there is actually this mass balance study available for a glacier in the Karakoram prior to 2000, although it only covers a 5 year period: Bhutiyani, M. R. 1999. Mass-balance studies on Siachen Glacier in the Nubra valley, Karakoram Himalaya, India. Journal of Glaciology, 45(149), 112-118.

P2, Fig. 1: it would be useful to label the location and names of some of the main peaks or towns in this region to make the map easier to follow. The lat/long labels around the margins are also currently too small to see.

P3, L11: the sentence 'The major advantage. . .' doesn't really make sense as written. Please reword.

P3, L13: please provide the resolution and spatial extent of the KH-9 imagery. I also think that you mean to refer to Table S1 here, not Table 1

P3, L16: should be 'database' (one word)

P31, L21: a few words to explain what the 'reseau grid' is would be useful as it's not a commonly used term. I also think that it should be spelled 'Réseau grid'

P31, L23: change 'GCPs have been collected. . .' to 'GCPs were collected. . .'. Also describe how and where the GCPs were chosen – e.g., Were they located on bedrock areas? How many were used? Were they chosen across a range of elevations?

P5, L5-6: I'm unclear as to why a 5% uncertainty was chosen for the glacier area map-

ping. If there are good optical satellite images available for this area, then presumably it should be relatively straightforward to map the glacier areas with <5% uncertainty?

P6, Fig. 2: I find the labels and dots on these figures quite difficult to see as they're so small. Also please indicate the source of information for identifying which glaciers are surge-type. As mentioned above, it would also be very useful to produce a DEM difference map for the entire study period (1973-2009)

P7, Table 1: similar to the comment for Fig. 2, please include a column to show the glacier mass balance values for the entire study period 1973-2009.

P7, L11: 'where lower' should be 'were lower'

P8, L1-3: the first sentence in this para is difficult to follow. The rest of this para is also quite awkwardly worded, with quite a few typos. Please be sure to check carefully. E.g., L8: change 'allowed to detect the surge activity' to 'allowed for detection of surge activity'. Also – what does 'south exposed glacier' mean? Do you mean southerly facing?

P8, Fig. 3: also show the total change from 1973-2009

P9, L7-9: it would be useful to add a few words here (or elsewhere) about the relatively rapid surge periodicity in the Karakoram: i.e., that within a 40 year period it's likely that you've captured a large part of a surge cycle (or even more than one). This is different to locations such as Svalbard, where the active and quiescent phases are typically much longer.

Figure S1: it's unclear as to which dates refer to which areas, particularly for the 1999-2009 image. Colour coding the date label and associated box would help

Table S1: indicate what (P/R, K/J) indicate in the header for column 3
* * *

---

## Author Comment (AC1) · 12 Dec 2016

We'd like to thank the reviewer for the constructive review. Please find below our reply to the comments. We will provide a more detailed reply along with the revised manuscript.

RC: This study utilizes digital elevation models extracted from satellite imagery (historical Hexagon and modern SRTM and ASTER) to calculate a regional geodetic mass balance for glaciers in the Hunza River basin, Karakoram region, using DEM differencing. The authors show that given the uncertainties of the methodology, the regional geodetic mass balance is not statistically different from zero change, consistent with previous mass balance studies on shorter more recent timescales. Their results suggest that the so-called "Karakoram anomaly" is not limited to the past _15 years, but

extends back to at least 1973. This is the first study using elevation differences to confirm this finding over a several-decade timespan, which supports previous studies showing no significant changes in debris cover or glacier area in the Karakoram over similar (1970's-present) time periods. Overall, it is a nice paper, and is ready for publication after a few minor additions.

Reply: Thank you.

RC: A table of values showing the standard deviation of mean elevation change between the ASTER and SRTM DEMs for assumed stable (non-glacier) terrain is needed to better assess the relative vertical accuracy of the DEMs.

Reply: We included an established uncertainty assessment for estimate the accuracy of the results. These results are assigned as uncertainty range to each resultant number and given in table1. We will include the information about the standard deviation of the elevation difference between the final ASTER DEM and the SRTM DEM, and also the differences between in the KH-9 and ASTER and KH-9 and SRTM DEMs.

RC: Regarding the satellite imagery datasets, a paragraph, table, or figure to clearly show which DEMs are being subtracted from one another for each given time period, i.e. SRTM minus Hexagon for 1973-1999, and ASTER minus SRTM for 1999-2009. This would serve the clarify the methods section significantly.

Reply: There is a figure in the supplementary material which shows the boundaries of the utilized images. There is no overlap of the Hexagon DEMs, hence it is clear which area was substracted. We agree, however, that this is not so straight forward for the ASTER DEMs. In case an area is covered by more than one DEM, we used the mean of all available DEMs. We will clarify in the revised manuscript.

RC: Since the primary motivation of the paper is to extend the geodetic mass balance record further back in time, I would recommend an additional calculation of the full timespan (1973 - 2009) mass balance. This would also serve to validate the 1973-

1999 and 1999-2009 mass balances, and remove the significant uncertainty regarding SRTM penetration into the ice.

Reply: We have done so for few glaciers to clarify that the results agree well to the results of the individual periods. We agree, however, that it would be beneficial to also include the DEM difference for the entire period. This will therefore be done for the revised manuscript.

RC: The equations used for estimating uncertainty lean toward the more conservative side (i.e. large error bars). For example, linearly adding up the errors in Eq. 3 instead of adding in quadrature, which assumes that the error components in Eq. 3 are completely correlated with one another. The authors should make clear in the conclusion of the manuscript - results show no statistical difference from zero change, given the somewhat large/conservative uncertainties used with the DEM differencing method.

Reply: We understand the concern, but think the uncertainty is more realistic when adding the uncertainties of the radar penetration, volume to mass conversion etc. and considering the error propagation would underestimate the uncertainty from our point of view. This is especially as the approach by Gardelle et al. 2013 provides rather conservative error estimates. However, even with smaller uncertainty estimates the general statement that there we no significant differences between the period before and after 2000 holds true. We will discuss this issue in the respective section and mention it also in the conclusions.

Specific comments:

P3 L6 Were any glaciers covered only partially by scenes from different years? If so, it may be best to use a weighted mean (weighted by the percentage of a glacier's area covered by each scene).

Reply: Yes, especially the larger glaciers are covered by more than one scene. We will used the mean value for the overlapping parts. We will use now the weighted mean of

all the scenes used for a glaciers considering the area coverage as suggested.

P3 L8 It is still somewhat unclear to me how the Cartosat-1 data is being used. I assume the authors compute Cartosat minus SRTM, then compare to ASTER minus SRTM in order to check consistency between the datasets. This should be further clarified in the text.

Reply: Yes, it was done to compare to the ASTER derived which have significant lower resolution. We will write: "Two high-resolution Cartosat-1 stereo scenes captured on 11 July 2010 (Table 1) were used to compare and investigate the consistency of the results gained with the lower resolution ASTER DTM."

P2 L18 "assuming a full penetration of the radar beam into snow..." - regarding the ablation region. What about additional penetration into the ice itself, is this taken into account?

Reply: Yes this is taken into account. We applied the correction suggested by Kääb et al. (2012) who analysed the penetration depth for a similar region.

P2 L20 It would be useful here to refer to the later section (3.2) so the reader can easily find the discussion regarding void filling with the ASTER GDEM2 and associated uncertainties.

Reply: We will refer to the section 3.2. as suggested.

P3 L24 "All stereo images have been processed with a RMS of < _1.5 pixels." Which aspect of the stereo photogrammetry is this referring to? Is this the reprojection error of triangulated ground control points after bundle adjustment, or something to do with the reseau grid distortion removal, or something else? A more detailed explanation is needed to interpret the meaning. P3 L28 See previous comment regarding P3 L24

Reply: We refer to the RMSE of the GCPs after triangulation. We will clarify in the revised manuscript and improve the caption of Table S2 accordingly.

[Figure]

P3 L32 What kind of spatial trend corrections were made? Rotation, translation, or perhaps polynomial surface corrections... if so are they first order (linear), or higher order polynomials, or some other method?

Reply: We applied a first order trend correction. We will clarify this in the revised manuscript.

P4 L12 Was the outlier threshold applied to both Hexagon and ASTER data, or to Hexagon only? If no outlier filtering was needed for the ASTER DEMs, this should be stated explicitly in the text. P4 L17 It would be helpful to know the percentage of total pixels excluded (using the outlier threshold filter) for each glacier, to ensure that no large regions were interpolated using the ordinary kriging; otherwise unrealistic elevations could result. The text later discusses the percentage of voids in the SRTM data, but says nothing regarding the percentage of data gaps in the Hexagon data. P6 L13 Going back to the previous comments regarding P4 L12 and P4 L17 – Since both time periods use different data sources and therefore contain differing amounts of data voids - a percentage of voids for each would help eliminate doubts regarding direct comparisons between the two time periods, which use SRTM - Hexagon for 1973-1999, and ASTER - SRTM for 1999-2009. Could a difference in data gaps/holes make a significant difference when comparing these datasets/time periods?

Reply: The outliers were filtered for all DEM differences, but it was more important for the DEM differencing using Hexagon data. We will clarify this in the revised manuscript and also provide the information about the resultant voids in the different data sets. The impact of the outlier filtering and the gap filling will be discussed in the discussion section.

P6 L11 "we confirm for the first time using elevation differences..." Should specify: over this longer time period (because elevation differences have been used over shorter time periods in previous studies).

Reply: We agree and will now specifically mention "for the period since 1973"."

P6 L17 What is meant by "different surge stages in the two periods..."? What is different between the two periods regarding surges, the magnitude, timing, or something else? Or is the word "different" simply being used in a fashion equivalent to "separate"?

Reply: We simply meant that, e.g. one period covered the active surge of the glacier while the second probably the quiescent phase. We will be now more explicit and provide few examples of selected glaciers.

P8 L1 When calculating the mass budget with the non-void-filled version of SRTM for comparison, were the voids interpolated, or was the mass balance computed using only the volume change of existing pixels, then divided by the glacier area only covered by existing pixels? More details would be helpful.

Reply: We will clarify and write: "Using the latter we calculated the surface elevation change for the existing pixels only. The resultant value of the mean surface elevation change for both periods differs only by about 0.02 m a-1"

Technical corrections:

P2 L7 complicate the

Corrected

P3 L10 "The major advantage of this dataset is besides the high spatial resolution and also the 12 bit pixel depth" – strange wording. Should change to something like: "The major advantages of this dataset are the high spatial resolution and 12-bit pixel depth."

Reply: The sentence will be rewritten to: "The major advantage of this dataset is besides the high spatial resolution the 12 bit radiometric resolution."

P7 L11 ": : : ASTER DEMs where lower: : :" change "where" to "were"

Reply: Corrected

P8 L6 voids

Reply: Corrected

P8 L4 versions

Reply: We think the singular is correct here: "We compared therefore the results of the void filled and the non-void filled version."

---

## Author Comment (AC3) · 12 Dec 2016

We'd like to thank the reviewer for the constructive review. Please find below our reply to the comments. We will provide a more detailed reply along with the revised manuscript.

RC: This is a useful study that provides the first long-term information (since the 1970s) about geodetic glacier mass balance for a region of the Karakoram. This is a region where many recent papers have suggested that glaciers are changing little in mass, but prior to this study little previous information has been available about glacier elevation changes before _2000. The techniques are well described, the errors are well quantified, and useful final conclusions are produced. Most of my comments are relatively minor and focused on technical issues, but there are two useful analyses that could be

undertaken that would help to strengthen the paper:

1. A computation of the total elevation changes over the period 1973-2009 should be completed. This would help to validate the patterns shown in the individual periods, potentially reduce the effect from individual surges, and provide evidence that mass balance has been stable over the long term.

Reply: We have done so for few glaciers to clarify that the results agree well to the results of the individual periods. We agree, however, that it would be beneficial to also include the DEM difference for the entire period. This will therefore be done for the revised manuscript.

RC: 2. Provide a plot and discussion of the change in geodetic mass balance with altitude for non surge-type glaciers. This could provide insight into whether changes are occurring at particular altitudes, even though the overall mass balance may be close to zero.

Reply: We agree that this information would be valuable. However, there are several other interesting analysis which could be done, e.g. comparison of the elevation change with altitude of debris-covered and non debris-covered glaciers. However, we chose the format of the Short Communication as we wanted to focus on the main new findings. In addition, we cannot add another figure due to the limitations in the chosen format.

Individual comments:

P2, L22: there is actually this mass balance study available for a glacier in the Karakoram prior to 2000, although it only covers a 5 year period: Bhutiyani, M. R. 1999. Mass-balance studies on Siachen Glacier in the Nubra valley, Karakoram Himalaya, India. Journal of Glaciology, 45(149), 112-118.

Reply: Fully agreed, Thank you for this correction. We will include: "The only exception is Siachen Glacier in eastern Karakoram for which Zaman and Liu (2015) corrected the

clearly negative value of 0.51 m w.e. a-1 by. Bhutiyani (1999) and estimated the mass budget to be between + 0.22 m and - 0.23 m w.e. a-1."

P2, Fig. 1: it would be useful to label the location and names of some of the main peaks or towns in this region to make the map easier to follow. The lat/long labels around the margins are also currently too small to see.

Reply: We agree. We will include the names of few larger villages and known mountain peaks and enlarge the font size of the coordinates

P3, L11: the sentence 'The major advantage: : :' doesn't really make sense as written. Please reword.

Reply: We agree and will write: "The major advantage of this dataset is besides the high spatial resolution the 12 bit radiometric resolution.

P3, L13: please provide the resolution and spatial extent of the KH-9 imagery. I also think that you mean to refer to Table S1 here, not Table 1

Reply: We agree and will include the information "…which has a ground resolution of about 8 m and a coverage of about 250 x 125 km…" and refer to table 1.

P3, L16: should be 'database' (one word)

Reply: corrected

P31, L21: a few words to explain what the 'reseau grid' is would be useful as it's not a commonly used term. I also think that it should be spelled 'Réseau grid'

Reply: We agree and will integrate few words.

P31, L23: change 'GCPs have been collected: : :' to 'GCPs were collected: : :'. Also describe how and where the GCPs were chosen – e.g., Were they located on bedrock areas? How many were used? Were they chosen across a range of elevations?

Reply: The numbers of the utilized GCPs were listed in Table S2. We will include the

following information in the manuscript: "GCP collection in rough terrain is challenging. Finally we were able to find 26/28 GCPs located at mountain peaks, large terrain features, and bridges which we distributed throughout the scenes and in different elevations as best as possible."

P5, L5-6: I'm unclear as to why a 5% uncertainty was chosen for the glacier area mapping. If there are good optical satellite images available for this area, then presumably it should be relatively straightforward to map the glacier areas with <5% uncertainty?

Reply: The glaciers are not that straightforward to map as there are several debris-covered ones. The major issue is, however, the correct delineation of the upper glacier boundary as several glaciers are avalanche fed and located below steep slopes where the boundary is not fully clear. Taken this into consideration and the fact that the study by Paul et al. (2013) revealed similar uncertainties in a mapping experiment where different experts provided glacier outlines, we think this estimate of the uncertainty is reasonable.

P6, Fig. 2: I find the labels and dots on these figures quite difficult to see as they're so small. Also please indicate the source of information for identifying which glaciers are surge-type. As mentioned above, it would also be very useful to produce a DEM difference map for the entire study period (1973-2009).

Reply: We will increase the size of the labels and dots and include the information about how we identified the surge-type glaciers in the text of the manuscript. We will analyse the elevation difference of the entire study period and include the figure in the text.

P7, Table 1: similar to the comment for Fig. 2, please include a column to show the glacier mass balance values for the entire study period 1973-2009.

Reply: We will include also the mass balance values for entire study period in table 1.

P7, L11: 'where lower' should be 'were lower'

Interactive
comment

Reply: corrected

P8, L1-3: the first sentence in this para is difficult to follow. The rest of this para is also quite awkwardly worded, with quite a few typos. Please be sure to check carefully. E.g., L8: change 'allowed to detect the surge activity' to 'allowed for detection of surge activity'. Also – what does 'south exposed glacier' mean? Do you mean southerly facing?

Reply: We agree that the paragraph is a bit difficult to understand. We will improve it in the revised manuscript.

P8, Fig. 3: also show the total change from 1973-2009

Reply: We will do.

P9, L7-9: it would be useful to add a few words here (or elsewhere) about the relatively rapid surge periodicity in the Karakoram: i.e., that within a 40 year period it's likely that you've captured a large part of a surge cycle (or even more than one). This is different to locations such as Svalbard, where the active and quiescent phases are typically much longer.

Reply: We agree and will include the information in the revised manuscript.

Figure S1: it's unclear as to which dates refer to which areas, particularly for the 1999-2009 image. Colour coding the date label and associated box would help

Reply: In general, it would be possible to identify as the dates are placed in the middle of the polygons representing the scenes. We agree, however, that it is a bit difficult to understand and the suggestion to colour code the date level is very good and we will improve the figure accordingly.

Table S1: indicate what (P/R, K/J) indicate in the header for column 3

Reply: (P/R, K/J) are indeed confusing and not needed. We forgot to delete and will delete for the revised version.

---

## Editor Comment (EC1) · E. Berthier (Editor) · 13 Dec 2016

Dear Authors,

Thanks for answering point by point to the referees. I am now looking forward to receive a revised version of your Brief Communication, taking into account all referee's advices.

Best regards,
* * *

---

## Author Response (AR1)

**Dear Editor,**

**We have carefully revised the manuscript taking the reviewers comments into account. We followed most of the comments. In case we, disagree we explained why. The most important comment which were also raised by you was that we did not calculate the mass changes for the entire period 1973 – ca. 2009. We did now carefully and the results confirm our previous estimates. Please see our detailed reply below. We used track changes so the all changes in the manuscript can be easily identified. We hope the manuscript can now be accepted.**

**With best regards,**

**Tobias Bolch**

**Reply to comments from Reviewer #1**

The comments are repeated in *italic*, our reply is given in **bold**.

*This study utilizes digital elevation models extracted from satellite imagery (historical Hexagon and modern SRTM and ASTER) to calculate a regional geodetic mass balance for glaciers in the Hunza River basin, Karakoram region, using DEM differencing. The authors show that given the uncertainties of the methodology, the regional geodetic mass balance is not statistically different from zero change, consistent with previous mass balance studies on shorter more recent timescales. Their results suggest that the so-called "Karakoram anomaly" is not limited to the past _15 years, but extends back to at least 1973. This is the first study using elevation differences to confirm this finding over a several-decade timespan, which supports previous studies showing no significant changes in debris cover or glacier area in the Karakoram over similar (1970's-present) time periods. Overall, it is a nice paper, and is ready for publication after a few minor additions.*

**Reply: Thank you.**

A table of values showing the standard deviation of mean elevation change between the ASTER and SRTM DEMs for assumed stable (non-glacier) terrain is needed to better assess the relative vertical accuracy of the DEMs.

**Reply: We included an established uncertainty assessment for estimate the accuracy of the results. These results are assigned as uncertainty range to each resultant number and given in Table 1. We now also include information about the standard deviation of the elevation difference between the final ASTER DEM and the SRTM DEM, and also the differences between in the KH-9 and ASTER and KH-9 and SRTM DEMs in the manuscript.**

Regarding the satellite imagery datasets, a paragraph, table, or figure to clearly show which DEMs are being subtracted from one another for each given time period, i.e. SRTM minus Hexagon for 1973-1999, and ASTER minus SRTM for 1999-2009. This would serve the clarify the methods section significantly.

**Reply: There is a figure in the supplementary material which shows the boundaries of the utilized images. There is no overlap of the Hexagon DEMs, hence it is clear which area was subtracted. We agree, however, that this is not so straight forward for the ASTER DEMs. In case an area is covered by more than one DEM, we used the mean of all available DEMs. We clarified this issue in the manuscript and refer now to a new table in the supplement. However, the new numbers only changed at the cm scale, which is well within the uncertainty.**

Since the primary motivation of the paper is to extend the geodetic mass balance record further back in time, I would recommend an additional calculation of the full timespan (1973 - 2009) mass balance. This would also serve to validate the 1973-1999 and 1999-2009 mass balances, and remove the significant uncertainty regarding SRTM penetration into the ice.

**Reply: We have done so for few glaciers to clarify that the results agree well to the results of the individual periods. We agree, however, that it would be beneficial to also include the DEM difference for the entire period. This is now done for the revised manuscript. In general, the values fit very well so our general statement is still supported by the analysis. However, we detected some deviation for Barpu Glacier. We investigated the potential reason more in detail and found unrealistic high elevation gain in parts of the accumulation region. We therefore adjusted the filtering slightly and the resultant mass balance value changed from -0.03 ± 0.18 to -0.10 ± 0.18. The value 1973-2009 is lower but still falls within the uncertainty of both subperiods. Within our detailed revisions of the calculation of the values for the entire period we got also slightly but insignificantly different values for the individual glaciers. These differences stem mainly from new outlier filtering detailed in the revised manuscript.**

The equations used for estimating uncertainty lean toward the more conservative side (i.e. large error bars). For example, linearly adding up the errors in Eq. 3 instead of adding in quadrature, which assumes that the error components in Eq. 3 are completely correlated with one another. The authors should make clear in the conclusion of the manuscript - results show no statistical difference from zero change, given the somewhat large/conservative uncertainties used with the DEM differencing method.
**Reply: We understand the concern, but think the uncertainty is more realistic when adding the uncertainties of the radar penetration, volume to mass conversion etc. and considering the error propagation linearly would underestimate the uncertainty from our point of view. This is the same approach used by Gardelle et al. 2013 also provides rather conservative error estimates. However, even with smaller uncertainty estimates the general statement that there we no significant differences between the period before and after 2000 holds true.**

Specific comments:

P3 L6 Were any glaciers covered only partially by scenes from different years? If so, it may be best to use a weighted mean (weighted by the percentage of a glacier's area covered by each scene).

**Reply: Yes, especially the larger glaciers are covered by more than one scene. We originally used the mean value for the overlapping parts. We now use the weighted mean of all the scenes used for a glaciers considering the area coverage as suggested. However, as stated above the resultant numbers changed only in the third decimal digit which is well within the uncertainty.**

P3 L8 It is still somewhat unclear to me how the Cartosat-1 data is being used. I assume the authors compute Cartosat minus SRTM, then compare to ASTER minus SRTM in order to check consistency between the datasets. This should be further clarified in the text.

**Reply: Yes, it was done to compare to the ASTER derived which have significant lower resolution. We write now: "Two high-resolution Cartosat-1 stereo scenes captured on 11 July 2010 (Table 1) were used to compare and investigate the consistency of the results obtained with the lower resolution ASTER DTM."**

P2 L18 "assuming a full penetration of the radar beam into snow..." - regarding the ablation region. What about additional penetration into the ice itself, is this taken into account?
**Reply: Yes this is taken into account. We applied the correction suggested by Kääb et al. (2012) who analysed the penetration depth for a similar region.**

P2 L20 It would be useful here to refer to the later section (3.2) so the reader can easily find the discussion regarding void filling with the ASTER GDEM2 and associated uncertainties.

**Reply: We referred to section 3.2. as suggested.**

P3 L24 "All stereo images have been processed with a RMS of < _1.5 pixels." Which aspect of the stereo photogrammetry is this referring to? Is this the reprojection error of triangulated ground control points after bundle adjustment, or something to do with the reseau grid distortion removal, or something else? A more detailed explanation is needed to interpret the meaning.
P3 L28 See previous comment regarding P3 L24

**Reply: We refer to the RMSE of the GCPs after triangulation. We improved the caption of Table S2 accordingly.**

*P3 L32* What kind of spatial trend corrections were made? Rotation, translation, or perhaps polynomial surface corrections... if so are they first order (linear), or higher order polynomials, or some other method?

**Reply: We applied a first order trend correction. We will clarify this in the revised manuscript.**

P4 L12 Was the outlier threshold applied to both Hexagon and ASTER data, or to Hexagon only? If no outlier filtering was needed for the ASTER DEMs, this should be stated explicitly in the text.
P4 L17 It would be helpful to know the percentage of total pixels excluded (using the outlier threshold filter) for each glacier, to ensure that no large regions were interpolated using the

ordinary kriging; otherwise unrealistic elevations could result. The text later discusses the percentage of voids in the SRTM data, but says nothing regarding the percentage of data gaps in the Hexagon data.

P6 L13 Going back to the previous comments regarding P4 L12 and P4 L17 – Since both time periods use different data sources and therefore contain differing amounts of data voids - a percentage of voids for each would help eliminate doubts regarding direct comparisons between the two time periods, which use SRTM - Hexagon for 1973-1999, and ASTER - SRTM for 1999-2009. Could a difference in data gaps/holes make a significant difference when comparing these datasets/time periods?

**Reply: The outliers were filtered for all DEM differences, but it was more important for the DEM differencing using Hexagon data. We have clarified this in the revised manuscript and also provide the information about the resultant voids in the different data sets. The impact of the outlier filtering and the gap filling is now covered in the discussion section.**

P6 L11 "we confirm for the first time using elevation differences..." Should specify: over this longer time period (because elevation differences have been used over shorter time periods in previous studies).

**Reply: We agree and will now specifically mention "based on 1973 Hexagon and ~2009 ASTER DEMs"."**

P6 L17 What is meant by "different surge stages in the two periods..."? What is different between the two periods regarding surges, the magnitude, timing, or something else? Or is the word "different" simply being used in a fashion equivalent to "separate"?

**Reply: We simply meant that, e.g. one period covered the active surge of the glacier while the second probably the quiescent phase. We are now more specific and provide an example.**

P8 L1 When calculating the mass budget with the non-void-filled version of SRTM for comparison, were the voids interpolated, or was the mass balance computed using only the volume change of existing pixels, then divided by the glacier area only covered by existing pixels? More details would be helpful.

**Reply: We write now: "Using the latter we calculated the surface elevation change for the existing pixels only. The resultant value of the mean surface elevation change for both periods differs only by about 0.02 m a-1"**

*Technical corrections:*

P2 L7 complicate the

**Corrected**

P3 L10 "The major advantage of this dataset is besides the high spatial resolution and also the 12 bit pixel depth" – strange wording. Should change to something like: "The major advantages of this dataset are the high spatial resolution and 12-bit pixel depth."

**Reply: The sentence was rewritten to: "The major advantage of this dataset is besides the high spatial resolution the 12 bit radiometric resolution."**

P7 L11 ": : : ASTER DEMs where lower: : :" change "where" to "were"

**Reply: Corrected**

P8 L6 voids

**Reply: Corrected**

P8 L4 versions

**Reply: We think the singular is correct here: "We compared therefore the results of the void filled and the non-void filled version."**

**Reviewer #2**
**We'd like to thank the reviewer for the constructive review. Please find below our reply to the comments. We will provide a more detailed reply along with the revised manuscript.**

RC: This is a useful study that provides the first long-term information (since the 1970s) about geodetic glacier mass balance for a region of the Karakoram. This is a region where many recent papers have suggested that glaciers are changing little in mass, but prior to this study little previous information has been available about glacier elevation changes before _2000. The techniques are well described, the errors are well quantified, and useful final conclusions are produced.
Most of my comments are relatively minor and focused on technical issues, but there are two useful analyses that could be undertaken that would help to strengthen the paper:

1. A computation of the total elevation changes over the period 1973-2009 should be completed. This would help to validate the patterns shown in the individual periods, potentially reduce the effect from individual surges, and provide evidence that mass balance has been stable over the long term.

**Reply: As this point was also raised by Reviewer 1, we now examine the total elevation change from 1973 – 2009. In general, the total elevation change over the entire period matches that found for the individual periods, and so our general statement is supported by the analysis. However, we detected some deviation for Barpu Glacier. We investigated the potential reason more in detail and found unrealistic high elevation gain in parts of the accumulation region. We therefore adjusted the filtering slightly and the resultant mass balance value changed from -0.03 ± 0.18 to -0.10 ± 0.18. The value 1973-2009 is lower but fits well within the uncertainty to the sum of both subperiods.**

RC: 2. Provide a plot and discussion of the change in geodetic mass balance with altitude for non surge-type glaciers. This could provide insight into whether changes are occurring at particular altitudes, even though the overall mass balance may be close to zero.

**Reply: We agree that this information would be valuable. However, there are several other interesting analyses which could be done, e.g. comparison of the elevation change with altitude of debris-covered and non debris-covered glaciers. However, we chose the format of the Short Communication as we wanted to focus on the main new findings. In addition, we cannot add another figure due to the limitations in the chosen format.**

Individual comments:

*P2, L22: there is actually this mass balance study available for a glacier in the Karakoram prior to 2000, although it only covers a 5 year period: Bhutiyani, M. R. 1999. Mass-balance studies on Siachen Glacier in the Nubra valley, Karakoram Himalaya, India. Journal of Glaciology, 45(149), 112-118.*

**Reply: Fully agreed, Thank you for this correction.**

**We will include: "The only exception is Siachen Glacier in eastern Karakoram, for which Zaman and Liu (2015) corrected the erroneous value of -0.51 m w.e. a$^{-1}$ given by Bhutiyani (1999), and estimated the mass budget to be between +0.22 m and -0.23 m w.e. a$^{-1}$."**

*P2, Fig. 1: it would be useful to label the location and names of some of the main peaks or towns in this region to make the map easier to follow. The lat/long labels around the margins are also currently too small to see.*

**Reply: We agree. We will include the names of few larger villages and known mountain peaks and enlarge the font size of the coordinates**

P3, L11: the sentence 'The major advantage: : :' doesn't really make sense as written. Please reword.

**Reply: We agree and wrote now: "The major advantage of this dataset is besides the high spatial resolution the 12 bit radiometric resolution.**

P3, L13: please provide the resolution and spatial extent of the KH-9 imagery. I also think that you mean to refer to Table S1 here, not Table 1

**Reply: We agree and included the information "…which has a ground resolution of about 8 m and a coverage of about 250 x 125 km…" and refer to table 1.**

P3, L16: should be 'database' (one word)

**Reply: corrected**

P31, L21: a few words to explain what the 'reseau grid' is would be useful as it's not a commonly used term. I also think that it should be spelled 'Réseau grid'

**Reply: We agree and wrote now:
"Image pre-processing, comprising the elimination of internal distortions based on the réseau crosses regularly distributed crosses on the image which have this purpose to be able correct effects of film distortion) and their removal thereafter, …"**

P31, L23: change 'GCPs have been collected: : :' to 'GCPs were collected: : :'. Also describe how and where the GCPs were chosen – e.g., Were they located on bedrock areas? How many were used? Were they chosen across a range of elevations?

**Reply: The numbers of the utilized GCPs were listed in Table S2. We included the following information in the manuscript: "GCP collection in rough terrain is challenging. Finally we were able to find 26/28 GCPs located at mountain peaks, large terrain features, and bridges which we distributed throughout the scenes and in different elevations as best as possible."**

P5, L5-6: I'm unclear as to why a 5% uncertainty was chosen for the glacier area mapping. If there are good optical satellite images available for this area, then presumably it should be relatively straightforward to map the glacier areas with <5% uncertainty?

**Reply: The glaciers are not that straightforward to map as there are several debris-covered ones. The major issue is, however, the correct delineation of the upper glacier boundary as several glaciers are avalanche fed and located below steep slopes where the boundary is not fully clear. Taken this into consideration and the fact that the study by Paul et al. (2013) revealed similar uncertainties in a mapping experiment where different experts provided glacier outlines, we think this estimate of the uncertainty is reasonable.**

P6, Fig. 2: I find the labels and dots on these figures quite difficult to see as they're so small. Also please indicate the source of information for identifying which glaciers are surge-type. As mentioned above, it would also be very useful to produce a DEM difference map for the entire study period (1973-2009).

**Reply: We increased the size of the labels and dots in the new included figure for the period 1973-2009 and included the information about how we identified the surge-type glaciers in the text of the manuscript ("Overall, we identified 28 surge-type glaciers (including 5 tributaries) based on the DTM differencing results in combination with morphological features like looped moraines or heavily crevassed glacier surfaces (Fig. 2)."). We analysed the elevation difference of the entire study period and include the figure in the text.**

P7, Table 1: similar to the comment for Fig. 2, please include a column to show the glacier mass balance values for the entire study period 1973-2009.

**Reply: We included also the mass balance values for entire study period in table 1.**

P7, L11: 'where lower' should be 'were lower'

**Reply: corrected**

P8, L1-3: the first sentence in this para is difficult to follow. The rest of this para is also quite awkwardly worded, with quite a few typos. Please be sure to check carefully. E.g., L8: change 'allowed to detect the surge activity' to 'allowed for detection of surge activity'. Also – what does 'south exposed glacier' mean? Do you mean southerly facing?

**Reply: We agree that some wording were a bit awkward. We improve it in the revised manuscript.**

P8, Fig. 3: also show the total change from 1973-2009

**We think most interesting for the profile is the difference between the two periods and showing the profiles will not prove important additional information for the selected glaciers. However, it the reviewer insists it is no problem to add. But we will then probably move this figure to the supplement as the length of the manuscript and number of figures is restricted in a short communication.**

P9, L7-9: it would be useful to add a few words here (or elsewhere) about the relatively rapid surge periodicity in the Karakoram: i.e., that within a 40 year period it's likely that you've

captured a large part of a surge cycle (or even more than one). This is different to locations such as Svalbard, where the active and quiescent phases are typically much longer.

**Reply: This is a good point. We include this information now and write "…most of the surge events should be covered by our study period of almost 40 years as the surge periodicity in the Karakoram is rather short with averages between ~25 and 40 years (Copland et al. 2011)."**

Figure S1: it's unclear as to which dates refer to which areas, particularly for the 1999-2009 image. Colour coding the date label and associated box would help

**Reply: We tried different possibilities to improve clarity, but also colour coding was not really better as several colours would be needed and they would then interfere with the DT difference image. It is possible to identify the dates as they are placed in the middle of the polygons representing the scenes. Therefor we would prefer to leave the figure as it is.**

Table S1: indicate what (P/R, K/J) indicate in the header for column 3

**Reply: (P/R, K/J) are indeed confusing and not needed. We forgot to delete and will delete for the revised version.**

[revised manuscript text omitted]

---

## Author Response (AR2)

Dear Etienne,

Thank you for your careful read. We have carefully revised the manuscript taking your comments into account and provide below a point to point reply. We used track changes so the all changes in the manuscript can be easily identified. We hope the manuscript can now be accepted.

With best regards,

Tobias Bolch on behalf of all authors

**Reply to comments:**
The comments are repeated in *italic*, our reply is given in **bold**.

*Comments in the decision letter:*
*Supplement. Table S3. "Dates ASTER DTMs used" --> "Dates ASTER DTMs"*
*"(wighted mean" --> "(Weighted mean)*
**Corrected**

*P8, Fig. 3 "also show the total change from 1973-2009". If there is an added value to include the 1973-2009 elevation change you should add it. And I think there is definitively one. Indeed, this point makes me think that you need to clearly state earlier in the manuscript that the correction for penetration has been applied as a mean average (right? the mean value from Kaab et al. [2012]). If I understood correctly, you did not correct for variations of penetration as a function of altitude. Should be clearly stated. Then, this is a good reason to show the 1973-2009 dh value in Figure 3 because the values for subperiods including SRTM may be slightly biased because of higher penetration at high elevation than in the lower part.*
**Yes, we applied the penetration correction as a mean average and state this now in the revised text. We also added the profiles of the elevation changes for the entire period.**

*Annotated comments:*
*P1, comment 1: Replaced by something like "but only one mass budget analysis" to take into account the publication by Zhou et al.*
**We write now: "only very few", since there is also the data for Siachen Glaciers and maybe there is a (non-peer-reviewed) study, which we might have missed.**

*P1, comment 2: My advice would be to finish with an implication sentence rather than this technical point about the data used. To what extent your study will help other scientists*
**We present now the data used earlier in the abstract. There are several implications (such as for hydrological modelling, mass balance modelling understanding surge-type glacier) and we do not want to highlight a specific one.**

*P1, comment 3: I think Rankl et al. 2014 could be cited here also.*
**Reference cited as suggested.**

*P1, comment 4: Revised in view of the Zhou et al. JOG paper*
**We write now: „However, almost no mass budget analyses are available for Karakoram glaciers prior to the year 2000 are rare. While this paper was in open discussion,, Zhou et al. (2017) reported mass budgets −0.09 ± 0.03 m w.e. a-1 of for the period ~1970 to 2000. For Siachen Glacier in eastern Karakoram…**

*P1, comment 5: measurements, available since*
**We think that "measurements that are available since 1961" is more appropriate.**

*P2, comment 6: maybe add in the legend the source for the basin boundary, I do not think it was provided elsewhere.*
**The boundary was generated by ourselves by GIS analysis and crosschecked with a boundary for a study by Silvan Ragettli. We do not think it is really needed to add this information in the figure caption. However, we can write if you'd prefer.**

*P3, comment 1: always a space between number and unit*
**Space included between number and unit. We checked also the entire manuscript but did not find any other missing space.**

*P3, comment 2: what about the season of acquisition. Was it a selection criteria? If yes, as I suspect, then mention it.*
**Of course this was a selection criteria and thought it was clear from „minimum snow cover". However, we agree that this is not so obvious and include now "close to end of ablation season" as a selection criteria.**

*P3, comment 3: why difference? This is simply the mean of the dates of the two scenes right?*
**Yes, we use the mean of the time differences between the two acquisitions. Details are of the dates and the time difference are given in Table S3. We moved the sentence now to the section on the prostprocessing (see also your comment 2 on page 4.)**

*P3, comment 4: I suggest starting a new paragraph here*
**New paragraph started.**

*P4, comment 1: this statement is not clear to me. Is it the value of the horizontal shift that you found using NK2011 and that your applied?*
**Yes; we write now: "The final horizontal shift…"**

*P4, comment 2: not 100% clear to me. How the weight are applied. Do you consider each glacier separataly or work pixel by pixel?*
**We agree that the procedure was not 100% clearly described. We write now:
"To calculate surface elevation changes, we subtract each older DTM from more recent DTM, and mosaic the difference grids to facilitate processing. Where the ASTER DTMs from different time periods overlapped, we calculate a weighted mean elevation change based on the time of acquisition and glacier coverage (Figure S1, Table S3)."**

*P5, 5 comment 1: poor wording I think. Fourth author should check...*
**We improved the wording and write now: „The standard deviation of the non-glacierized terrain can serve as a first estimate of the uncertainty and is…"**

*P5, comment 2: move Delta_H to subscript*
**Corrected.**

*P6, comment: If you look carefully at their map you will discover many artefacts in their accumulation areas... coregistration was not perfect in this study. I do not want you to elaborate on this in the paper (not his scope) but just that you are aware of this*
**Thank you for the information.**

*P6, comment 2: other could argue that DLR deliver freely some TDX DEMs. Not really relevant, at least at this location in the paper*
**We removed "freely available".**

*P7, comment 1: little has been done to incorporate in the text the results for the 1974-2009 time period, although it was a strong comment of all reviewers. Maybe a few words are needed. Suggest a slight mass loss rather than balanced budget no?*
**We agree and address now more specific the entire study period. We also agree that results hint to on average to slight mass loss although the results are insignificant. We changes the statements accordingly and also adjusted the title to:**
**"Nearly balanced glaciers in the Hunza catchment (Karakoram) since the 1970s"**

*P8, comment 1: to be updated including Zhou et al. JOG*
**Updated accordingly.**

*P9, comment 1: Maurer*
**Corrected.**

*P9, comment 2: for the whole basin? For what period? 1973-2000?*
**Yes, for the whole basin and the period 1973 – 2009. This information is now included.**

*P10, comment 1: for what period?*
**A different penetration correction would affect both periods. We hence write now "±".**

*P10: comment 2: add "together with Zhou et al. (2017)" or something similar*
*P10: comment 3: The 1973-2009 mass balance suggests a slight mass loss...*
**We usually do not include references in the conclusion. We rewrote the conclusions and focus now more on our study region and the investigation on surge-type glaciers.**

[revised manuscript text omitted]

---

## Author Response (AR3)

Dear Editor,

Thank you for your additional careful read. We agree that the new statements are not really backed-up by the data and were not formulated in a way that they are easy to understand. We therefore adjusted text and formulate the sentences now a bit more carefully. See our reply below.

We are looking forward to your final decision,

Tobias

**Point 1**

*8.20. "It is therefore important to cover the entire surge cycle of surge-type glaciers in order to have valid estimates of their mass budgets." I am not sure this is the best formulation. One can have a reliable estimate of the glacier-wide mass balance even if only the active/quiescent part of the surge cycle is observed. The mass balance will only be representative of this specific period, not the whole surge cycle. Do the authors want to imply here that the mass balance could be different depending on the phase of surge cycle? Do they have some data and physical processes to back up this possibility? (could be that a more crevassed glacier surface after the active phase is more prone to melt or that due to the advance of the glacier beyond the earlier front during the active phase an overall lowering of the hypsometry is observed and thus stronger melt can occur…).*

We wanted to make an important point that the mass budget of a surge-type glacier may differ depending on the period covered. This was more a discussion and not really quantified with the data. We formulate the text now a bit more carefully. See the track changes document on page 8.

*The reservations expressed above can also be made concerning two new sentences in the conclusion. "Especially surge-type glaciers show different elevation change characteristics and more negative mass budgets in the second period. During a surge, ice is transported rapidly from an upper reservoir zone to the ablation region and is prone to melt at the lower elevations." First, from Table1, the mass budgets for the two categories (surge-type/non surge-type) are not statistically different (error bounds). Then your second sentence seems to understate that the mass budget would be more negative during the active phase than during the quiescent phase but in table 1 you did not discriminate "surge-type glaciers" that were active (surging) and those that were quiescent. And how can one isolate the effect on the mass balance of the climate/stage in the surge cycle.*

We agree that our formulation was not the best and that the differences are not significant. We omitted the respective statement in the conclusions. We added on sentence that the geodetic mass budget based on KH-) and ASTER Data confirm the results from the individual periods without having the uncertain radar penetration. See the track changes document on page 12.

*Title: The mass budget of these glaciers is in balance (or nearly). However, given their strongly unstable dynamics (e.g., surge) can you really state that the "glaciers are balanced"? The same apply to L13 of the abstract.*

The term "balanced" refers to balanced budgets. We think this is clear from the context and we wrote already in the first submission "that glaciers in the Hunza River basin (Central Karakoram) were

on average in balance since the 1970s" and also our first title "Glaciers in the Hunza Catchment (Karakoram) are in balance since the 1970s" was not questioned by the reviewers. We think our sentence in the abstract "that glaciers in the Hunza River basin (Central Karakoram) were on average in balance or showed slight insignificant mass loss within the period ~1973 – 2009." can therefore remain like it is. We use now the old title but slightly adjusted it to "
[revised manuscript text omitted]